# Effect of TiO_2_ Nanoparticles and Extrusion Process on the Physicochemical Properties of Biodegradable and Active Cassava Starch Nanocomposites

**DOI:** 10.3390/polym15030535

**Published:** 2023-01-20

**Authors:** Carolina Iacovone, Federico Yulita, Daniel Cerini, Daniel Peña, Roberto Candal, Silvia Goyanes, Lía I. Pietrasanta, Lucas Guz, Lucía Famá

**Affiliations:** 1Laboratorio de Polímeros y Materiales Compuestos (LPMC), Departamento de Física, Facultad de Ciencias Exactas y Naturales, Universidad de Buenos Aires, Buenos Aires C1428EGA, Argentina; 2Instituto de Investigación e Ingeniería Ambiental, Escuela de Hábitat y Sostenibilidad, Campus Miguelete, Universidad Nacional de San Martín, San Martín 1650, Provincia de Buenos Aires, Argentina; 3Instituto de Física de Buenos Aires (IFIBA-CONICET), Facultad de Ciencias Exactas y Naturales, Universidad de Buenos Aires, Buenos Aires C1428EGA, Argentina; 4Centro de Microscopías Avanzadas y Departamento de Física, Facultad de Ciencias Exactas y Naturales, Universidad de Buenos Aires, Buenos Aires C1428EGA, Argentina

**Keywords:** starch nanocomposites, TiO_2_NP, extrusion, tensile strength, UV-shielding, industrial composting

## Abstract

Biodegradable polymers have been strongly recognized as an alternative to replace traditional petrochemical plastics, which have become a global problem due to their long persistence in the environment. In this work, the effect of the addition of titanium dioxide nanoparticles (TiO_2_NP) on the morphology, physicochemical properties and biodegradation under industrial composting conditions of cassava starch-based nanocomposites obtained by extrusion at different screw speeds (80 and 120 rpm) were investigated. Films performed at 120 rpm (S_120_ and S_120_-TiO_2_NP) showed completely processed starch and homogeneously distributed nanoparticles, leading to much more flexible nanocomposites than those obtained at 80 rpm. The incorporation of TiO_2_NP led to an increase in storage modulus of all films and, in the case of S_120_-TiO_2_NP, to higher strain at break values. From the Kohlrausch–Williams–Watts theoretical model (KWW), an increase in the relaxation time of the nanocomposites was observed due to a decrease in the number of polymer chains involved in the relaxation process. Additionally, S_120_-TiO_2_NP showed effective protection against UV light, greater hydrophobicity and faster biodegradation in compost, resulting in a promising material for food packaging applications.

## 1. Introduction

Plastics are one of the main sources of solid waste in large cities. In particular, the consumption of single-use plastics from food packaging has been identified as a global environmental pollution crisis since they can take up to hundreds of years to decompose [1,2,3]. Compostable biopolymers offer an efficient solution to plastic pollution since some of them are able to fully biodegrade under adequate environmental conditions [4]. Under composting conditions, polymer biodegradation takes place mainly in two stages. First, fragmentation of polymer chains occurs due to the hydrolysis process initialized by chemical or biological reactions. Then, microorganisms turn the carbon from polymer chains and the oxygen from the air into carbon dioxide (CO_2_), water and biomass [5]. Biodegradability can be evaluated by quantifying the concentration of CO_2_ evolved. According to ISO 17088 standards, one of the conditions that plastics must meet to be considered biodegradable under industrial composting conditions is to reach 90% degradation in less than 6 months [6].

The need to reduce the excessive use of single-use plastics that damage the environment generated a great demand for new biodegradable and/or compostable materials with specific properties and easy production. The capabilities of many types of nanocomposites based on biodegradable polymers are currently being explored to try to meet this demand [7,8,9]. Starch has been one of the most attractive polymers for years to achieve new developments for the packaging industry due to its high capacity to form low-cost and gluten-free biodegradable films [10,11], and because it comes from renewable and highly accessible sources [12,13].

Among processing methods, extrusion is a technique that has been extensively used to produce plastic materials. However, starch extrusion involves multiple chemical and physical reactions, such as gelatinization, decomposition, melting and crystallization [14,15]. Different investigations over the years have shown that the use of glycerol as a plasticizer favors the gelatinization of starch when the water content is limited, as occurs with the extrusion technique [16]. Additionally, shear stress and processing temperature contribute to gelatinization and are relevant parameters to consider in starch extrusion [17]. Recently, Vedove et al. showed the absence of starch granules and broken grains on the surface and fracture surface of cassava starch-glycerol films prepared by extrusion using a screw speed of 80 rpm. González-Seligra et al. showed that the use of 80 rpm and 120 rpm in the extrusion of cassava starch with 20 wt.% of glycerol was very promising to obtain homogeneous thermoplastic starch films [18,19].

The addition of nanofillers, such as silver, zinc oxide and titanium dioxide nanoparticles, has been widely used in science to obtain starch-based nanocomposites for external packaging due to their outstanding properties and antimicrobial characteristics [20,21,22]. Particularly, titanium dioxide nanoparticles (TiO_2_NP) have shown bactericidal capacity against *E. coli* and *S. aureus* and the ability to reflect and absorb UVA (320–400 nm) and UVB (290–320 nm) rays, protecting against sunburn, photoaging and extending product’s shelf life after harvesting [23,24,25]. In addition, FDA has a restriction of titanium dioxide of 1 wt.% when it is used as a food additive [26].

Regarding starch/TiO_2_NP, Arezoo et al. and Goudarzi et al. stated that these fillers can efficiently protect products against UV light, being useful as UV-shielding packaging materials [27,28]. Yousef et al. showed that the incorporation of TiO_2_NP into thermoplastic starch films prepared by casting led to more flexible materials with higher mechanical resistance while Oleyaei et al. reported a slight increase in the hydrophobicity of potato starch with 1 wt.% of TiO_2_NP nanocomposites [29,30].

However, to date, there are few investigations on the effect of TiO_2_NP on thermoplastic starch films produced by an industrial technique such as extrusion. Xiong et al. used different concentrations of TiO_2_NP in films from potato starch (60%) with glycerol (40%) prepared by blending and obtained nanoparticle agglomeration [31]. However, the authors observed a decrease in the moisture content and an increase in the hydrophobicity, crystallinity and elongation at break of the starch-based nanocomposites due to the incorporation of the nanoparticles. To the best of our knowledge, there are no studies on the compostability under adequate standards of starch materials with TiO_2_NP.

This work aims to evaluate the effect of the incorporation of TiO_2_NP on the morphology, crystalline structure, mechanical properties, UV-vis absorption and biodegradability under industrial composting conditions of cassava starch-based nanocomposites obtained by extrusion and thermo-compression technology at different processing conditions.

## 2. Materials and Methods

### 2.1. Materials

Cassava starch with about 18 wt.% amylose was obtained by C.A.I.S.A. (Costa Rica, Misiones, Argentina), analytical grade glycerol and titanium dioxide nanoparticles (TiO_2_NP, mixture of rutile and anatase, purity > 99%) were purchased in Sigma-Aldrich (Merck KGaA, Darmstadt, Germany). TiO_2_NP diameter sizes were smaller than 100 nm, according to the manufacturer.

### 2.2. Films Preparation

Thermoplastic starch films were developed using cassava starch (60%), glycerol (20%) and distilled water (20%) with and without TiO_2_NP (1 wt.%) using extrusion at 80 rpm or 120 rpm of screw speed and thermo-compression (defined as S_80_, S_80_-TiO_2_NP, S_120_ and S_120_-TiO_2_NP). These parameters were chosen from previous studies, in which starch-glycerol films not very sticky and very easy to handle were achieved [32]. A co-rotating twin screw extruder (Nanjing Kerke Extrusion equipment Co., Ltd. Jiangsu, China) was employed with a screw diameter of 16 mm and a length-to-diameter ratio of L/D = 40. The extruder has ten independently controlled temperature zones in the barrel and a 4 mm cylindrical aperture at the end. The temperature profile employed was set at 80 °C/100 °C/110 °C/120 °C/130 °C/140 °C/140 °C/140 °C/130 °C/120 °C for both screw speeds. This temperature profile was chosen considering that in previous works a lower temperature profile was employed at 80 rpm or 120 rpm, which led to some broken starch granules [32,33]. From extrusion process a large thread was obtained that was stabilized in a desiccator at room temperature and relative humidity (RH) of 57% given by a saturated solution of sodium bromide (NaBr). Then, pieces of the thread (~4 g) were used to form films from a thermostatic hydraulic press following Ochoa-Yepes et al. procedure [34]. Samples were placed in between two Teflon sheets in the press and heated at 130 °C for 5 min. Afterward, the pressure was increased up to 1 kPa for 15 min and then to 5 kPa for another 15 min. Finally, the temperature was lowered to 40 °C keeping the pressure at 5 kPa. The resulting films were measured 10 times per system, and all the films resulted of 0.30 ± 0.03 mm in thickness. Films were stored again at 57% RH for 14 days in order to avoid starch retrogradation and evaluate stabilized samples until characterization [35,36,37].

### 2.3. Films Characterization

#### 2.3.1. FTIR

ATR-FTIR spectra of all the developed film were recorded in a Jasco FT-IR 4100 spectrometer (Hachioji, Japan) with attenuated total reflectance (ATR) in the range 4000−600 cm^−1^ with a resolution of 0.4 cm^−1^ and an average of 64 scans. Experiments were carried out in triplicate and no significant differences were observed.

#### 2.3.2. Scanning Electron Microscopy (SEM)

In order to investigate the cross-section morphology of the films, SEM analysis was performed using field emission scanning electron microscopy (FEG-SEM SUPRA40, Overkochen, Alemania). Samples were fractured under liquid nitrogen and then coated with platinum by sputtering before observation.

#### 2.3.3. X-ray Diffraction (XRD)

XRD patterns of starch films were evaluated by a PANalytical Empyrean (Malvern Panalytical, Malvern, UK) using Cu (K-α) radiation (λ =1.54 Å, 40 kV and 30 mA). Samples were scanned over a diffraction angle range of 2*θ* = 5–35° at a step size of 0.026° and a scan speed of 200 s. A smooth curve connecting the peak baselines was plotted on the diffractograms, and the relative crystallinity was estimated by the ratio of the area above the smooth curve (upper diffraction peak area) over the total diffraction area [38,39].

#### 2.3.4. Water Vapor Permeability (WVP)

Water vapor permeability (WVP) of developed films was determined according to ASTM E-96 procedure with the following modifications [33,40]: Film samples were placed in acrylic permeation cells, with an exposed circular area of 3.8 × 10^−4^ m^2^, previously filled with CaCl_2_. Cells were conditioned in desiccators at RH of 75% (NaCl) and stored at room temperature to simulate a humidity-controlled chamber. The sealed cell was weighed every 24 h for 7 days. Linear models of the results were performed, and the slope *G* (g/s) was obtained. Then WVP (g/msPa) was calculated using Equation (1):(1)$$WVP=\frac{Ge}{A\;DP\;HR}$$
where *e* (m) is the film thickness, *A* (m^2^) is the test area, *DP* (Pa) is the saturation vapor pressure of water at room temperature, and *HR* is the relation of the difference in relative humidity between the cell and the desiccator.

Three replicates were performed for each sample. The mean value and standard deviation are reported.

#### 2.3.5. Moisture Content (MC)

Moisture content (*MC*) was determined in accordance with the standard method of the AOAC [41]. Samples of each system were initially weighted (*m_i_*∼0.5 g), dried in an oven at 100 °C for 24 h and then weighted (*m_f_*). Moisture content was calculated as Equation (2):(2)$$MC=\frac{m_{i}-m_{f}}{m_{i}}\;\times \;100.$$

Assays were carried out in triplicate.

#### 2.3.6. Contact Angle (θ)

The contact angle (θ) was reported as an indicator of the hydrophobicity of the materials. It was determined using an Optical Tensiometer (OneAttension theta—Biolin Scientific, Stockholm, Sweden) at 25 °C, from the angle formed by the tangent of a water drop (2 μL) deposited on the surface of the film and the film surface. Tests were performed in triplicate.

#### 2.3.7. Uniaxial Tensile Properties

Uniaxial tensile properties were studied using Brookfield CT3 Texture Analyzer (Middleborough, MA, USA) at room temperature according to ASTM D882-02 (2002) recommendations at a crosshead speed of 4 mm/min and room temperature [42]. Films with known thickness were cut in strips of 5 mm width and clamped between two jaws with an initial distance of 35 mm Stress-strain curves were recorded and the Young’s modulus (E), stress at break (σ_b_), strain at break (ε_b_) and tensile toughness (T) values were obtained from them. Young’s modulus was calculated as the slope of the stress-strain curves in the initial linear region and tensile toughness was determined as the area under the curves up to the point of fracture. At least, 10 replicates were performed for each system.

#### 2.3.8. Mathematical Modeling of KWW

The stress (σ)–strain (ε) curves can be analyzed to consider them as a spectroscopy technique [43] from estimating the average relaxation time <***τ***> through a known mathematical model, the semi-empirical function Kohlrausch–Williams–Watts (KWW). This model presents the stress σ = σ(ε) for viscoelastic materials as Equation (3):(3)$$\sigma \left(\varepsilon \right)=E_{0}\varepsilon +E_{1}{\left(k\tau^{*}\right)}^{\beta }\left(1-e^{-{\left(\frac{\varepsilon }{k\tau^{*}}\right)}^{\beta }}\right)$$
where k is the strain rate of the test, *E*_0_ is the modulus of the relaxed state (for long times), *E*_0_ + *E*_1_ is the initial modulus and ***β*** is an empirical parameter that takes values between 0 and 1 and quantifies the width of the spectrum of relaxation times. The characteristic time ***τ**** is an average relaxation time that characterizes molecular motions in amorphous systems [44], and can be expressed as ***τ**** = <***τ***> × ***β***. 

The KWW model on the films was evaluated from the fit of σ-ε curves with Equation (3), from which the parameters ***τ**** and ***β*** were determined.

#### 2.3.9. UV-Vis

UV–vis spectra was studied employing a spectrophotometer (Shimadzu UV-1800, Tokio, Japan). Film samples (1 × 3 cm) were cut and put into a quartz cell, and the absorbance spectrum between 250 and 900 nm was registered. All tests were studied in triplicate presenting no significant differences.

#### 2.3.10. Biodegradability

A Direct Measurement Respirometric (DMR) system based on ISO 14855-1 set up was developed to evaluate the aerobic biodegradation of the films under industrial composting conditions [45]. The DMR system was designed to measure the evolved CO_2_ with a CO_2_ non-dispersive infrared gas analyzer (NDIR, model MH-Z19 from Zhengzhou Winsen Electronics Technology Co., Zhengzhou, China with a measurement range from 0 to 5000 ppm). DMR was composed of 4 parallel series (Figure 1) in which pressurized air circulates through soda lime to remove the CO_2_ present in the air. Then, the air is circulated through a chamber containing deionized water to moisten it, and then through the composting reactor, where the biodegradation process takes place. Afterward, the air is redirected through a water trap (a condenser unit) to remove water vapor from the outflow gas stream before reaching the NDIR sensor. The reactor content (compost and the studied materials) is the only component of the system that is modified between each line. The first line only contains compost (blank), the second a known reference material (native starch powder) and the other two the samples to be measured. In addition, a temperature-controlled system was made by covering the reactors with heating jackets and using a thermosensor to set the temperature at 58 °C, in order to simulate industrial composting conditions.

The NDIR was used to measure the concentration of CO_2_ present in the air outflow from the reactors. Since water condensed after exiting the environmental chamber can damage the NDIR, a water filter (Serie QBM1 G1/4′’, Micro Pneumatic S.A.S, Bogotá, Colombia) was installed.

To convert the concentration of CO_2_ to mass, Equation (4) was used [46]
(4)$$mCO_{2}=\frac{CFT44}{22,414\;\times \;10^{6}}\;\times \;100$$
where *mCO_2_* is the mass of evolved CO_2_ (g), *C* is the CO_2_ concentration (ppm), *F* is the flow rate, *T* is the time between each measuring sequence, 22,414 is the volume of 1 mol of gas in cm^3^ at STP, 44 is the molecular weight of carbon dioxide (g/mol), and 10^6^ is the ppm conversion factor.

The percentage of mineralization (biodegradation) can be determined following Equation (5) and Ref. [45]:(5)$$\%M=\frac{mCO_{2}-mCO_{2b}}{mCO_{2th}}\;\times \;100.$$
where *%M* is the percentage of mineralization, *mCO_2_* (g) is the amount of evolved CO_2_ in the sample, *mCO_2b_* (g) is the amount of evolved CO_2_ in the blank, and m*CO_2th_* (g) is the theoretical amount of carbon dioxide. The carbon content of the samples was 41%, 42% and 41.2% for starch, S_120_ and S_120_-TiO_2_NP, respectively, determined by Carbon Determination in Soil (SC832 Series Series, LECO, United States of America) and it was used to calculate *mCO_2th_*.

For this test, 250 g of industrial compost based on agricultural residues (Orgánicos de Argentina S.A.) with pH 6.8, volatile solids 18 ± 2% and 59 ± 2% humidity was used in each reactor. The compost was sieved on a 2 mm sieve and preconditioned at 58 °C for 3 days before using. Deionized water was incorporated to adjust the moisture content to about 50%. Twenty-five grams of perlite (Terrafertil S.A., Moreno, Argentina) was mixed with the compost and a 3 cm column of perlite was added at the bottom of the reactor to provide better aeration. Twenty grams of samples cut into 2 × 2 cm were mixed with the compost and perlite. On the other hand, native starch powder used as reference was mixed with the compost and perlite. In both cases, a proportion of 1:6 parts (on dry basis) compost was used, as ISO 14855-1:2012 recommends.

During the test, the reactors were incubated in the dark. The compost was stirred and compost humidity was adjusted every day. The CO_2_ evolved from the reactors was recorded every 6 h.

To observe the macroscopic physical changes occurring during composting conditions, three films of the 20 g of sample were enclosed in plastic grids and mixed with the rest of the samples, compost and perlite. At 1.5, 3 and 5 days of the test, samples were carefully removed from the grid and photographed. Then, they were taken back into the reactor in order to not interrupt the cumulative CO_2_ assessment.

#### 2.3.11. Statistical Analysis

Data analysis was made using Python with *p* < 0.05 de confidence. The reported values are the mean and the error of the mean.

## 3. Results and Discussion

### 3.1. FTIR

The main normalized ATR-FTIR spectra of different films are shown in Figure 2. All curves exhibited the typical behavior of thermoplastic starch-based materials [47,48], with a significant absorption peak at 3600−3000 cm^−1^ attributed to the O–H stretching vibration due to free and inter- and intra-molecular hydrogen bonding of the films [49]; a double peak at 2930 and 2850 cm^−1^ associated the symmetric and asymmetric C-H stretching; and a peak at 1640 cm^−1^ assigned to the water adsorbed by starch molecules, among others. There are no changes in the wave number of the bands nor the appearance of new bands due to the different processing or the nanoparticle’s addition. However, after the incorporation of TiO_2_NP, slight broadening in the 3300 cm^−1^ band was observed. This behavior suggests an increased degree of hydrogen bonding since oxygen atoms in TiO_2_ formed hydrogen bonding with H atoms of the OH groups in the starch [30,50].

### 3.2. Morphology

FE-SEM micrographs of the cryogenic fracture surface of the films showed some broken starch grains in the matrix made at 80 rpm (Figure 3a, marked with arrows, and Figure 3e), which seem to have fully processed with the use of 120 rpm (S_120_). This suggests that the higher mechanical energy produced by the increase in the screw speed contributed to the improvement in the complete process of the starch grains.

The incorporation of TiO_2_NP behaved markedly differently depending on the nanocomposite manufacturing process. While in the case of S_80_-TiO_2_NP an inhomogeneous distribution of nanoparticles was observed with the presence of some holes, in the nanocomposite prepared at 120 rpm a great improvement in the distribution of the nanofiller and prevention of holes generation was found (Figure 3b,d). In an image of S_120_-TiO_2_NP with higher magnification (Figure 3g,h), it can be seen nanoparticles agglomerations. Nevertheless, the agglomerates seem to be homogeneously dispersed (Figure 3b,g).

### 3.3. X-ray Diffraction

Figure 4 presents the X-ray diffraction patterns of the different samples. Diffraction peaks centered at 11.7°, 14.8°, 16.9° and 18° are related to A-type structure and those at 22.2° and 24.2° correspond to B-type structure [51]. These peaks are related to the retrogradation of starch during storage, although there could be a contribution from some non-gelatinized granules [19,51,52]. Peaks at 12.9° and 19.6° correspond to V_h_-type structure, related to the insertion of glycerol molecules within the starch helical structure [51].

Relevant changes in X-ray pattern and relative crystallinity (RC) of the films were observed when starch was processed at different rpm. The relative intensity of the peak at 18° decreased while that of the peaks corresponding to the Vh-type structure at 19.6° increased in S_120_ samples compared to S_80_ (Figure 4). These results are due to a better starch processing in S_120_ and thus to the possible greater interaction between starch and glycerol, suggesting that more glycerol molecules are within the helical structure of the starch in this film [33,53]. As a consequence, lower RC values were obtained for the films processed at 120 rpm (Table 1).

With TiO_2_NP incorporation, the characteristic peak of V_h_-type crystalline structure at 19.6° decrease in S_120_-TiO_2_NP, which demonstrates that the nanoparticles could have restricted the recrystallization of molecular starch, as has already been reported in the literature [31], since they may act as nucleation centers in a starch matrix [54]. Furthermore, a new peak at around 25.2° appears in the films containing the nanofillers, which is characteristic of TiO_2_ nanoparticles [31], and causes an increase in the relative crystallinity of the nanocomposites.

### 3.4. Water Vapor Permeability (WVP)

WVP values did not show significant differences between the films, as can be shown in Table 2. A trend towards an increase in WVP of S_80_ compared to S_120_ is observed, which could be due to the more tortuous path for water due to the presence of broken starch grains. The incorporation of TiO_2_NP decreases WVP of S_80_-TiO_2_NP and increases it in S_120_-TiO_2_NP. In the first case, the behavior could be associated with the presence of some holes in S_80_-TiO_2_NP that favored the diffusion of water molecules through the film, leading to increases in WVP. In the case of S_120_-TiO_2_NP, it would be due to the greater distribution of TiO_2_NP in this nanocomposite with respect to S_80_-TiO_2_NP. The presence of well-dispersed nanoparticles could increase the tortuous path by dispersing water molecules and thus decreasing the diffusion rate.

### 3.5. Moisture Content (MC)

Table 2 shows a slight decrease in the moisture content (MC) of S_80_ compared to S_120_ perhaps as a result of its higher crystallinity since, as it is known an increase in crystallinity leads to a decrease in MC [55,56]. The addition of the nanoparticles led to a slight decrease in MC in the case of S_120_-TiO_2_NP and was explained by the stable hydrogen bonds between TiO_2_NP and matrix components leading to a decrease in the number of water molecules with the possibility of being captured by starch. Thereby, the penetration of water molecules is more difficult, increasing the water resistance of the nanocomposites, as shown in the literature on starch-TiO_2_NP nanocomposites prepared by casting [28,30].

### 3.6. Contact Angle (θ)

As can be seen in Table 2, no significant differences in contact angle values were observed between S_80_ and S_120_, nor with the addition of TiO_2_NP. A decreasing trend of Θ in S_80_-TiO_2_NP can be observed compared to S_80_ (Figure 5), possibly due to the presence of some holes in this nanocomposite, which leads to an increase in the water drop absorption and thus in the hydrophilicity of the film. In contrast, in the case of S_120_-TiO_2_NP, a more hydrophobic film was obtained. This was possible as a consequence of the strong hydrogen bonding interaction between the nanoparticles and the matrix that left fewer OH groups available to interact with the water drop, as seen in MC and ATR-FTIR measurements [31].

### 3.7. Uniaxial Tensile Properties

Figure 6 shows the tensile stress (**σ**)–strain (***ε***) curves of all films, which exhibit the typical behavior for thermoplastic starch materials [57]: a linear elastic zone followed by a non-linear elastic zone until failure. The stress at break (***σ***_b_), strain at break (***ε***_b_), Young’s modulus (E) and toughness (T) values obtained from the ***σ***-***ε*** curves, as well as the parameters from the theoretical KWW model are shown in Table 3. Relevant differences were observed between the films produced at different rpm. S_80_ presented higher E and ***σ***_b_ values but lower strain at break than S_120_ and no significant differences in T. The increase in E was expected due to the presence of starch granules in S_80_, as shown in SEM images (Figure 3a), which led to a more crystalline material (as seen in XRD). The broken grains may have been the cause of early material failure due to faster crack propagation produced during the deformation.

The addition of TiO_2_NP showed different effects depending on the type of processing performed. In both cases, increases in E were obtained due to the higher modulus of TiO_2_NP compared to the starch matrix, being around 20% in S_120_-TiO_2_NP. The strain at break markedly decreased in S_80_-TiO_2_NP and increased in the nanocomposite prepared at 120 rpm probably due to the well-dispersed nanoparticles in S_120_-TiO_2_NP. It is not so common to find in the literature nanocomposites with increases in ***ε***_b_, since nanoparticles incorporation frequently generates early propagation of failures [30,58]. Increases in the strain at break were reported in starch-TiO_2_NP bionanocomposite prepared by blending [35] and in chitosan/starch blend-based films with 1 wt.% of TiO_2_NP [31,59]. These last authors showed that when NP presented heterogeneous dispersion in the polymeric matrix, decreases in **ε**b occurred, concluding that heterogeneity could act as stress concentrator.

Regarding KKW model, an increase in ***τ**** and a decrease in ***β*** with increasing extrusion speed. This was expected considering that when 120 rpm is used, the crystalline phase decreases due to the better processing of the starch grains, as previously discussed. With the addition of nanoparticles, a marked decrease in ***τ**** and a slight increase in ***β*** was obtained in the case of S_120_-TiO, suggesting that the nanoparticles tend to increase the number of polymeric molecules involved in the relaxation process, which was related to the highest elongation of this nanocomposite.

### 3.8. UV-Vis

Figure 7 shows the UV-vis absorbance spectra of the samples, where S_80_ was similar to S_120_. As can be observed, there was a drastic difference in the curves between the matrix and the nanocomposites. S_120_-TiO_2_NP and S_80_-TiO_2_NP presented an absorption peak at around 340 nm, which is not observed in the matrices. This phenomenon is explained by considering that TiO_2_NP has UV shielding properties and, consequently, the starch-nanocomposites containing this nanofiller should present good UV shielding ability [60]. UV shielding properties were observed in starch films containing 1 wt.% of TiO_2_NP prepared by casting [28,30]. S_120_-TiO_2_NP and S_80_-TiO_2_NP did not present significant differences, indicating that the screw speed during the material’s process does not change light absorption properties.

### 3.9. Biodegradability

Based on the improved properties of the materials produced at 120 rpm over those prepared at 80 rpm, biodegradability tests were performed on S_120_ and S_120_-TiO_2_NP. The percentage mineralization curves obtained from cumulative CO_2_ evolution as a function of composting time are presented in Figure 8.

During the first 10 days, the mineralization rate of starch (reference) was higher than that of S_120_, and afterward, both materials’ biodegradation rates were similar almost all of the time. S_120_ mineralization’s curve is similar, as shown in the literature in starch-based films prepared by extrusion, and its initial slower biodegradation could be a consequence of the lower surface area of the films compared to starch powder [61,62]. Briassoulis et al. observed similar behavior when comparing the biodegradation in soils of PHB [63]. In powder PHB, the authors observed a higher initial biodegradation rate than in 2 × 2 cm films. The fast mineralization curve of starch powder is possible because it has a higher surface area than films and it is easily available for microbial assimilation [64].

The incorporation of TiO_2_ nanoparticles noticeably accelerated the biodegradation rate and increased the final biodegradation percentage of S_120_-TiO_2_NP compared to S_120_ and the reference. This behavior has not been previously observed in starch nanocomposites. Luo et al., obtained higher biodegradation rates under industrial composting conditions of (Poly-lactic) acid films when incorporating TiO_2_NP [65], suggesting that this behavior could be related to hydrolytic degradation, since TiO_2_NP may accelerate some hydrolytic reactions. Similarly, Del Campo et al. observed that the incorporation of ZnO in a biodegradable commercial polymer (Ecovio) significantly accelerates the disintegration process under a composting environment [66,67,68], indicating that the nanoparticles could initiate or advance some hydrolytic reactions. It was expected that the incorporation of TiO_2_NP delayed the starch film biodegradation since it is well-known that these nanoparticles provide antimicrobial activity to polymeric nanocomposites [69,70]. However, in this work, antimicrobial activity was evaluated by the halo inhibition assay in agar plates and no activity was observed (see Appendix A). Water and microorganisms could damage the interface between the starch matrix and nanoparticles, increasing the possibility of hydrolysis, as Zamir et al. proposed from the higher degradation rate of PLA films when starch nanoparticles were incorporated [71].

At the end of the test, after 42 days, S_120_-TiO_2_NP, S_120,_ and powder starch reached 106 ± 5%, 83 ± 7% and 83 ± 6% biodegradation, respectively. S_120_-TiO_2_NP showed greater mineralization than 100%, which according to the literature is an indication of the priming effect attributed to the over-degradation of the indigenous organic carbon present in the compost when glucose and its polymers are tested [64,72,73,74]. According to ISO 14855-1 and the European Standard EN 13432, S_120_-TiO_2_NP can be identified as biodegradable since its percentage of mineralization was greater than 90% and that of the positive control (powder cassava starch) was greater than 70% [45,75].

Figure 9 shows the macroscopic appearance of the films at different times of disintegration in the DMR system. After only 1.5 days of testing, fragmentation was observed in both samples due to the absorption of water and the adhesion of organic matter on them. The films already started to break down in pieces mainly due to the fast hydrolytic degradation process. This behavior was more noticeable in S_120_-TiO_2_NP, where more pieces were disintegrated compared to S_120_. On the 3rd day, only a very few pieces of S_120_-TiO_2_NP were observed, while on the 5th day a total degradation was noticed and S_120_ still presented many pieces.

Even though the films had almost completely disappeared by the third day and fifth day, their biodegradation process had not been completed, as can be seen in Figure 8. This occurs because in the initial degradation steps the material fragments and forms microplastics that still contain organic carbon that could undergo microbial biodegradation and be converted into CO_2_. These results demonstrate the relevance of determining materials biodegradability by employing respirometric assays, since macroscopic analysis could underestimate the required time for the complete mineralization, especially in the case of fast-degrading polymers such as starch-based films.

The results of biodegradation tests show that TiO_2_NP significantly accelerates the disintegration process of starch-based nanocomposites under an industrial composting environment.

## 4. Conclusions

The investigation of biodegradable polymers as an alternative to replace traditional petrochemical plastics for environmental preservation is relevant nowadays. In the present study, the extrusion processing conditions of cassava starch films and the incorporation of TiO_2_NP in thermoplastic starch-based nanocomposites were evaluated. Increasing the extrusion screw speed from 80 rpm to 120 rpm led to materials with lower crystallinity that can withstand greater deformations before breaking and nanocomposites with TiO_2_NP homogeneously dispersed in the starch matrix. Based on the results, greater hydrophobicity in the nanocomposites prepared at 120 rpm was observed due to the destruction of the hydrogen bond formed between the O in H_2_O and the H in C-O-H of the starch by the addition of TiO_2_NP, which also led to higher Young’s modulus and relaxation time values. The characteristics of TiO_2_NP were reflected in the effectiveness of the UV light protection of the films for their possible use in light-blocking packaging and in their greater crystallinity. This investigation revealed that TiO_2_NP used in starch films initiated or advanced some hydrolytic reactions under industrial composting conditions, accelerating the disintegration and biodegradation of the material. It was concluded that the use of TiO_2_NP in cassava starch-based materials could be promising for its use as fast degrading material under industrial composting conditions and light-blocking packaging with improved mechanical resistance.

## Figures and Tables

**Figure 1 polymers-15-00535-f001:**
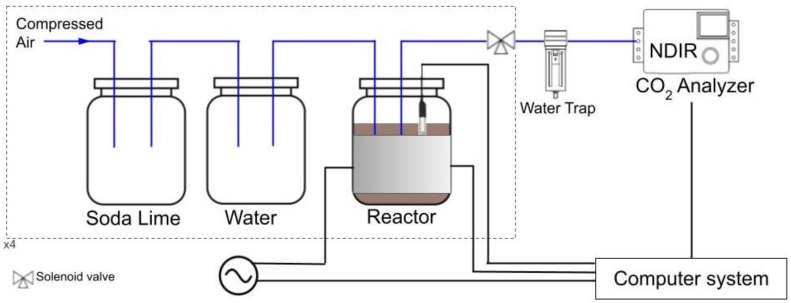
Schematic drawing of direct measurement respirometric system.

**Figure 2 polymers-15-00535-f002:**
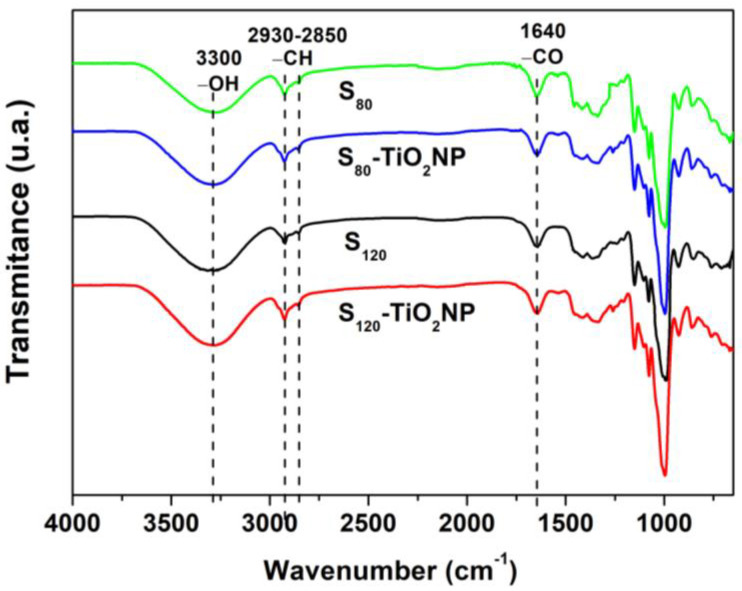
FTIR spectrum of the different studied films.

**Figure 3 polymers-15-00535-f003:**
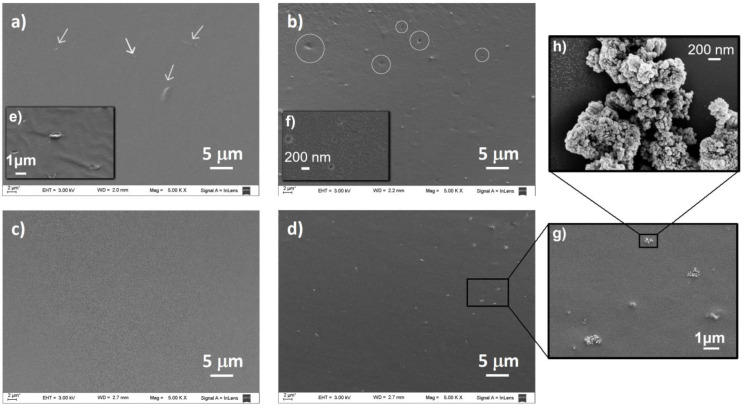
SEM micrographs of the cryogenic fracture surface of (**a**) S_80_, (**b**) S_80_-TiO_2_NP, (**c**) S_120_ and (**d**) S_120_-TiO_2_NP. Arrows indicate broken starch grains, which can be seen with higher magnification in (**e**), and circles indicate holes that can be seen with higher magnification in (**f**–**h**) show higher magnification of nanoparticle agglomeration.

**Figure 4 polymers-15-00535-f004:**
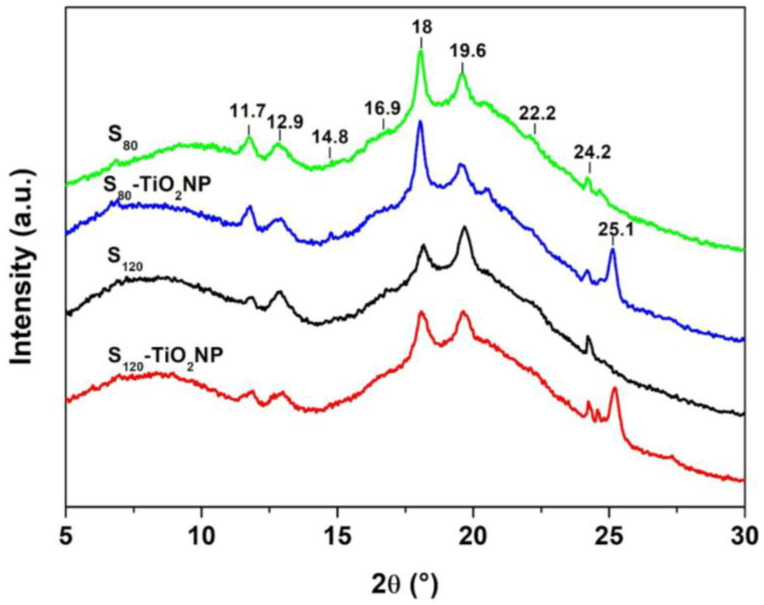
X-ray diffraction patterns of S_80_, S_80_-TiO_2_NP, S_120_, and S_120_-TiO_2_NP.

**Figure 5 polymers-15-00535-f005:**
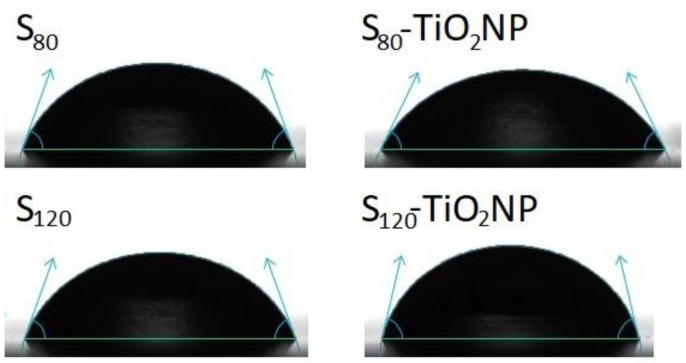
Image of the drop of distilled water deposited on the surface of the films.

**Figure 6 polymers-15-00535-f006:**
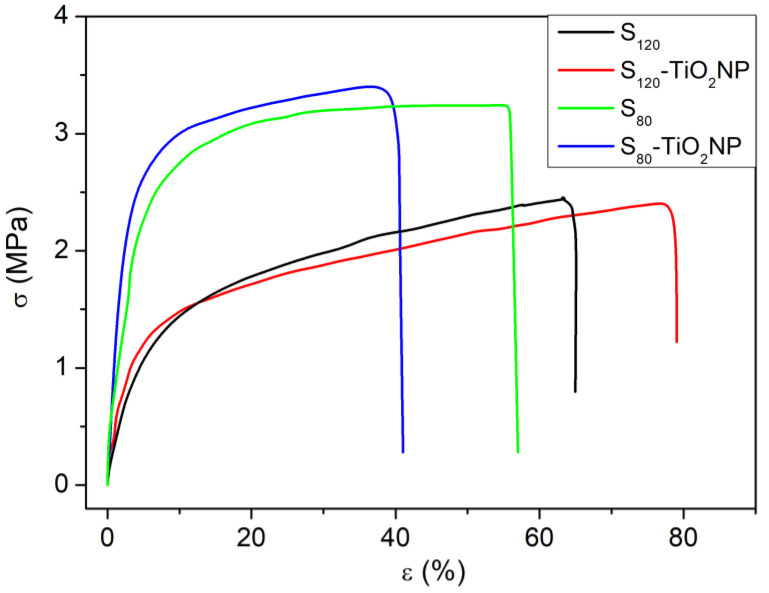
Tensile stress (**σ**)–strain (**ε**) curves of S_80_, S_80_-TiO_2_NP, S_120_ and S_120_-TiO_2_NP.

**Figure 7 polymers-15-00535-f007:**
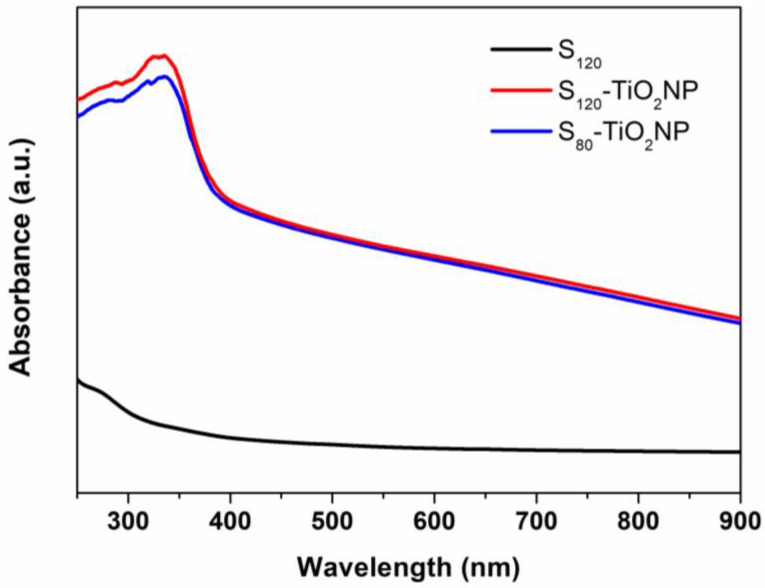
UV–vis absorption spectra of Starch and Starch/TiO_2_ nanocomposite films.

**Figure 8 polymers-15-00535-f008:**
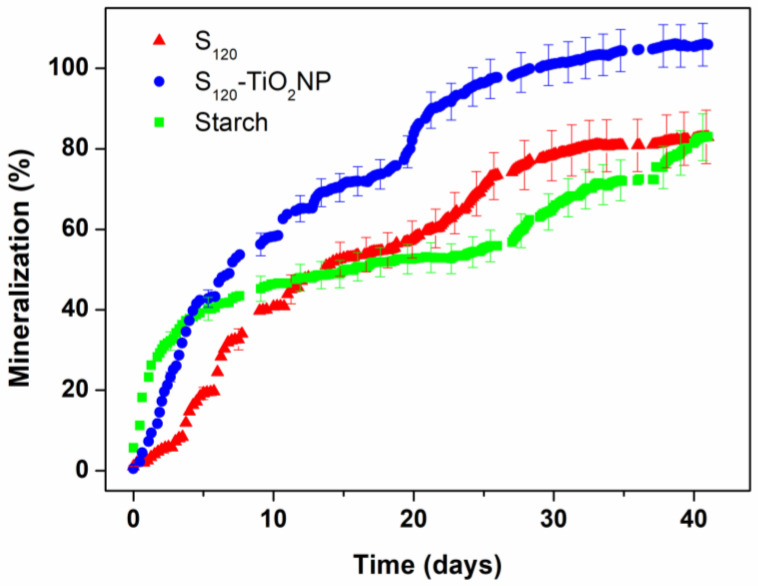
Percentage mineralization as a function of time for S_120_, S_120_-TiO_2_NP and starch in the DMR system. Mineralization error bars for the replicates are represented at selective times to facilitate the reading of the plots.

**Figure 9 polymers-15-00535-f009:**
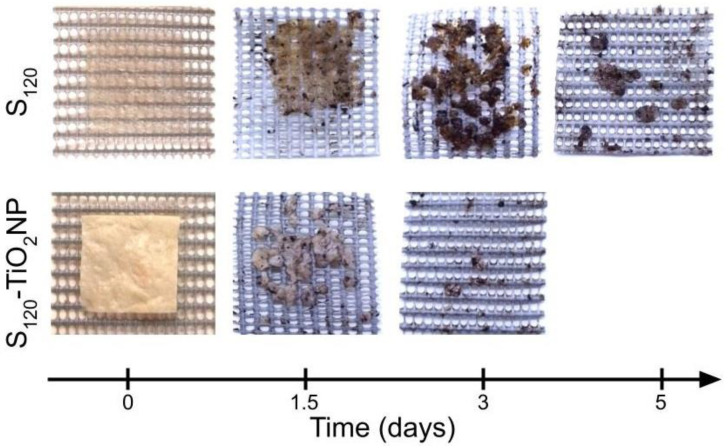
Photographs of the tested materials at different timings of the degradation process.

**Table 1 polymers-15-00535-t001:** Relative crystallinity of the films obtained from the X-ray diffraction patterns.

Peaks (°)	Type	Relative Crystallinity (%) [±0.2]			
		S_80_	S_80_-TiO_2_NP	S_120_	S_120_-TiO_2_NP
18	A	2.7	3.7	1.7 ^a^	2.0 ^a^
19.6	V_h_	2.0 ^a^	1.9 ^a^	2.7	2.0 ^a^
25.1	Titanium	0	3.3 ^a^	0	3.4 ^a^
Total	A-B-V_h_	8.2	12.5	7.4	10.5

^a^ Values with the same letters in the files are not significantly different (*p* > 0.05).

**Table 2 polymers-15-00535-t002:** Results of water vapor permeability (WVP), moisture content (MC) and contact angle (θ) of the developed materials.

	WVP	MC (%) [±1]	Θ (°)
S_80_	1.1 ± 0.1 ^a^	18 ^a^	70 ± 3 ^a^
S_120_	1.3 ± 0.3 ^a^	22	70 ± 3 ^a^
S_80_-TiO_2_NP	1.4 ± 0.4 ^a^	18 ^a^	64 ± 4 ^a^
S_120_-TiO_2_NP	0.9 ± 0.1 ^a^	19 ^a^	77 ± 3

^a^ Values with the same letters in the columns are not significantly different (*p* > 0.05).

**Table 3 polymers-15-00535-t003:** Results of the Young’s modulus (E), stress at break (***σ***_b_), strain at break (***ε***_b_), tensile toughness (T), and the parameters from the theoretical Kohlrausch–Williams–Watts (KWW) model (***τ**** and ***β***).

	*E* (MPa)	*σ_b_* (MPa)	*ε_b_* (%) [±4]	*T* (MJ/m^3^)	*τ**	*β*
S_80_	81.5 ± 0.8	3.2 ± 0.3 ^a^	57	1.6 ± 0.2 ^a^	26.28 ± 0.03	0.820 ± 0.001
S_120_	39.6 ± 0.7	2.7 ± 0.2 ^a,b^	66	1.3 ± 0.1 ^a,b^	178 ± 2	0.511 ± 0.003
S_80_-TiO_2_NP	87.3 ± 0.8	3.4 ± 0.3 ^a^	41	1.2 ± 0.1 ^b^	28.63 ± 0.07	1.000 ± 0.003
S_120_-TiO_2_NP	48.5 ± 0.6	2.5 ± 0.3 ^b^	79	1.5 ± 0.5 ^a,b^	32.08 ± 0.3	0.634 ± 0.001

^a,b^ Values with the same letters are not significantly different (*p* > 0.05).

## Data Availability

Not applicable.

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
