# Peer review of "Effect of TiO2 Nanoparticles and Extrusion Process on the Physicochemical Properties of Biodegradable and Active Cassava Starch Nanocomposites"

_polymers, 2023, doi:10.3390/polym15030535_

Round 1

Reviewer 1 Report

The manuscript "Effect of TiO2 nanoparticles and extrusion process on the physicochemical properties of biodegradable and active cassava starch nanocomposites" submitted by Iacovone et al. to Polymers deals with the study of the effect of the addition of titanium dioxide nanoparticles (TiO2NP) on the morphology, physicochemical properties, and biodegradation under industrial composting conditions of cassava starch-based nanocomposites obtained by extrusion at different screw speeds. Although the manuscript is well written and the topic could be of interest, the results and discussion presented do not support their conclusions, and therefore the manuscript should be rejected.

Some general comments:

- Although two different screw speeds were used in the extruder (80 and 120 rpm), nothing is mentioned about why these conditions were used and the effect they were looking for? What happens if different speeds are used? During the extrusion, different temperatures were used; why this temperature profile was used? Why the authors used 60% Cassava and 20% Glycerol? It is known that different glycerol content drastically affects the physical properties.

-(lines 98-99)

It can be read: "The resulting films of 0.30 ± 0.03 mm in thickness were stored again at 57% RH for 14 days until characterization."

Why? What happens if the characterization is performed immediately or over a more extended time?

How was the thickness controlled?

-(lines 112-113)

It can be read: "Water vapor permeability (WVP) of developed films was determined according to 112 ASTM E-96 procedure with modifications [23,24]."

What are the modifications?

-(lines 167-171)

It can be read: "Then, the air circulates through a chamber containing deionized water to humidify the air, and then into the composting reactor, where the biodegradation process takes place. Lastly, the air circulates through a water trap in order to remove moisture, and, then, a non-dispersive infrared gas analyzer measures the concentration of the CO2 evolved from the reactor."

How much humidity can get in the air?

Is there a way to estimate the humidity content?

What is the purpose of adding and then removing the humidity?

-(line 192)

It can be read: "…%M is the percentage of mineralization…”

It is not clear the concept of mineralization. From the text, one can deduce mineralization is inversely proportional to biodegradation. But what is it?

-(lines 228-230)

It can be read: "…who argued that it could be due to a possible hydrogen 228 bond interaction formed between the hydroxyl groups of the starch and the OH of the nanoparticles [32]."

Where does the OH in TiO2 nanoparticles come from?

- Figure 3. SEM

If the TiO2 particles are incorporated in the nanometer size, then a micrograph in the micrometer range is inadequate.

-(lines 256-257)

It can be read: "The behavior in the latter case could be associated with the presence of holes and cracks in S80-TiO2NP that favored the difusión of water molecules through the film…"

Where are those holes and cracks?

-(lines 262-264)

It can be read: "This same effect, but due to the presence of broken starch grains 262 would be occurring in the matrix made at 80 rpm, thus a decrease in WVP compared to S80-TiO2NP would be expected."

Do you have broken and unbroken starch grains in your films?

-(lines 289-292)

It can be read: "The result is consistent considering that the material produced at 80 rpm exhibited a higher density of cracks and holes when TiO2NP were incorporated, which leads to an increase in the water drop absorption..:"

Where are those holes and cracks?

-(lines 289-292)

It can be read: "The materials processed at 304 80 rpm presented higher E and ?b values but lower strain at break than S120, maintaining the toughness without significant changes. This is probably due to the presence of micrometric starch granules that failed to gelatinize during the heating phase..:"

Which result supports this analysis?

Author Response

We appreciate the time dedicated to the evaluation of our work and we thank the comments and suggestions from the reviewer in order to improve the manuscript. We incorporated new results and discussions to enrich the manuscript and to better support the conclusions. All the changes were marked in red in the new version of the manuscript. For convenience of reading, the answers to the reviewer's questions are in the attached file.
We hope that the revised manuscript addresses all of the reviewer’s questions.

Thank you very much.

Sincerely yours,

Dr. Famá

Reviewer 2 Report

The manuscript reports effect of TiO2 nanoparticles and extrusion process on the physicochemical properties of biodegradable and active cassava starch nanocomposites. This manuscript will be considered for acceptable after revised based on the following comments:

1.     The samples were prepared with or without TiO2NP (1 wt.%). How about other content of TiO2NP, above or below 1 wt.%? To reveal the effect of TiO2NP on nanocomposites, the authors are encouraged to add more experiments.

2.     The authors mentioned the presence of holes and cracks in S80-TiO2NP. However, from the SEM image, the size of holes and cracks are the same with that of TiO2NP. Did the holes and cracks occur due to the process of cryo-fracturing?

3.     The authors mentioned that this is probably due to the presence of micro-metric starch granules that failed to gelatinize during the heating phase in S80 as it was shown by SEM (Figure 2a), which is more crystalline than the matrix leading to a remarkable increase in E with respect to S120. However, from Figure 2a, the micro-metric starch granules were not clearly observed. In addition, XRD or DSC should be performed to investigate the crystalline structure.

4.     For better understanding the mechanism of biodegradation, the variation of molecular chain structure should be analyzed, e.g., GPC.

5.     There are some typos in this manuscript, e.g., in line 103 “cm−1”, line 244 “S80-TiO2NP”, Table 1 “θ”. The authors are encouraged to double check the manuscript and correct all the typos to make the manuscript more readable.

Author Response

We thank the reviewer for the time dedicated to the evaluation of our work. We considered their comments in order to improve the quality of the manuscript. All the changes were marked in red in the new version of the manuscript. For convenience of reading, the answers to the reviewer's questions are in the attached file.
We hope that the revised manuscript addresses all of the reviewer’s questions.

Thank you very much.

Sincerely yours,

Dr. Famá

Reviewer 3 Report

1) English should be improved.

2) Check all format. For example, -1 should be in superscript in line 103.

3) I suggest polarizing microscope and DSC should be used to analyze the crystalline structure

Author Response

(The authors gave the same response as above.)

Reviewer 4 Report

The manuscript "Effect of TiO2 nanoparticles and extrusion process on the physicochemical properties of biodegradable and active cassava starch nanocomposites" deals with the use of starch and glycerol, with and without TiO2 NP, in order to fabricate biodegradable films by extrusion evaluating the effect of rpm. The manuscript, in general is organized and logical. Nevertheless, some key points must be reviewed as follows.

1) Section 3.1. FTIR. The statement regarding the change of intensity could be indicative of TiO2-starch interaction is not correct since the ATR-FTIR methodology is not a quantitative one. In addition, the potential interaction could be seen by a slight deformation of -OH related peaks (not a shift), but you say that there is not any deformation, then review this and amend accordingly.

2) Section 3.1 FTIR. Please explain why the signals associated to methylene (-CH2-) are almost disappear in spectrum corresponding to S120 sample. Also, please give the functional groups-wave number association in Figure 2 or in a Table. 

3) Figure 3 (b & d). The images show something on the surface, probably TiOparticles, indicating nanoparticles agglomeration and non-efficient dispersion in the starch. Please provide elemental analysis of such points as well as images with higher magnifications to appreciate differences in the samples (morphology) at scales corresponding to SEM.

4) Section 3.3. The presented discussion could be supported by morphology of samples obtained by SEM at nanoscale images. Please provide images considering this.

5) Section 3.4. The statement regarding the sorption capacity of broken grains must be reviewed since broken grains would give higher surface area with higher capacity to sorb liquids. 

6) Section 3.4, lines 275-277. Please provide the potential mechanism of interaction between TiO2 and methylene groups, as you mention that it could be happen.

7) Section 3.5. Please provide the contact angle value for the pristine TiO2 since discussion would be different if you consider it. In addition, please provide more fundamental discussion considering the potential interaction between the component (provide chemical schema regarding this).

8) Please provide a typical stress-strain profile of each sample, and mention how the toughness was evaluated. 

9) Section 3.7. TiO2 NP are able to generate free radicals under solar light. Thus, the degradation rate also can be accelerated by the photodegradation mechanism, what do you think about it?

10) Conclusion. This section must be rewritten considering that the current one is more a description of some experiments and results but there is not a conclusion, which explain what are the results attributed to. 

11) The abstract must be reviewed since it mentions that S120 samples are more flexible than those obtained at 80 rpm, but evaluation of this has not been presented nor discussed.

Author Response

(The authors gave the same response as above.)

Round 2

Reviewer 1 Report

Although no answer to the reviewers was found, the authors modified the manuscript according to the observations given. Therefore, this version is suitable for publication.

Reviewer 2 Report

The authors have made substantial revisions on the manuscript according to the comments from the reviewers. My questions are well addressed and i'd like to recommend the manuscript to publish.

Reviewer 4 Report

Thank you for considering the observations and addressed them. The revised manuscript looks much better than the previous one.